# THE CROSSWORD PUZZLE: SIMPLIFYING DEEP NEURAL NETWORK PRUNING WITH FABULOUS COORDINATES

## ABSTRACT

Pruning is a promising technique to shrink the size of Deep Neural Network models with only negligible accuracy overheads. Recent efforts rely on experience-derived metric to guide pruning procedure, which heavily saddles with the effective generalization of pruning methods. We propose The Cross Puzzle, a new method to simplify this procedure by automatically deriving pruning metrics. The key insight behind our method is that: *For Deep Neural Network Models, a Pruning-friendly Distribution of model's weights can be obtained, given a proper Coordinate*. We experimentally confirm the above insight, and denote the new Coordinate as the Fabulous Coordinates. Our quantitative evaluation results show that: the Crossword Puzzle can find a simple yet effective metric, which outperforms the state-of-the-art pruning methods by delivering no accuracy degradation on ResNet-56 (CIFAR-10)/-101 (ImageNet), while the pruning rate is raised to 70%/50% for the respective models.

## 1 INTRODUCTION

Pruning Deep Neural Network models is promising the reduce the size of these models while keeping the same level of the accuracy. Prior arts focus on the designs of the pruning method, such as iterative pruning (Han et al. (2015a), one-shot pruning (Lee et al. (2018)), pruning without training (Ramanujan et al. (2020)), etc. However, prior works craft the pruning metrics via additional efforts, based on the testing experiences.

Our goal in this work is to design a method for automatically searching a proper metric for model pruning. Based on the classic pipelines (e.g. Genetic Algorithm (Mitchell (1998)) and Ant Colony Optimization (Dorigo & Di Caro (1999)), we first systematically summarize such a method requires three components: ❶ Basic building blocks of pruning criteria; ❷ Objective function to evaluate auto-generated pruning metrics; ❸ Heuristic searching process to guide the searching. Based on the above summary, prior works mainly focus on the first and third components (for instance, we can use $L_1$-norm (Li et al. (2016)) and geometric median (He et al. (2018b)) as building blocks, and simulated annealing (Kirkpatrick et al. (1983)) as our searching guider). Therefore, it's still unclear that how objective functions should be measured for the quality of a certain pruning metric (namely the unfilled letters in our "crossword puzzle" denotation).

This motivates us to examine the essential condition(s) of a good-quality pruning criterion. Based on a simple magnitude-based pruning method (Han et al. (2015b)) and the follow-up weight distribution analysis (Liu et al. (2018)), we formalize that one essential condition and describe it as follows:

Given a coordinate $\Psi$ (the formal expression of a pruning criterion) and neural network model $M$, $\Psi$ is highly likely to be high-qualified[1], if the distribution $D(M)$ got from $\Psi(M)$ obeys the following requirements:

- **Centralized distribution:** the statistics are concentrated on one center in the distribution, which is an important symbol of overparameterized neural networks.

---

[1]We refer "a coordinate to be highly-qualified", if we can use it to prune neural network model with (almost) no accuracy drop under a relatively-high pruning rate.

- **Retraining recovers centralized distribution:** through retraining, statistics can regather at the original distribution center after the peak on that center is cutting off by pruning, which means that the objects located at the center are able to replace each other.

- **Central collapse:** at the end of the pruning, we can observe that retraining can't drive the statistics to fill the center void again (namely central collapse). This is a signal demonstrating that there is nearly none redundancy in the model, which also alludes to the success of our criterion selection.

We denote such a coordinate $\Psi$ as the **Fabulous Coordinates**, and the distribution $D$ generated by it as the **Fabulous Distribution**. Based on our formalization, we can convert the *pruning criterion searching problem* into finding the Fabulous Coordinates. By quantitatively depicting the Fabulous Distribution using Loose-KL-Centralization-Degree (LKL-CD), we formulate the objective function and build the **Crossword Puzzle**, a pruning criteria searching framework.

We experimentally evaluate the effectiveness of the Crossword Puzzle. First, we use the Crossword Puzzle to find a Fabulous Coordinate. Then, we leverage the found Fabulous Coordinate to guide our pruning, and the results confirm the effectiveness of our method on CIFAR-10 and ImageNet. Our results show that we can prune VGG-16-bn (CIFAR-10)/ResNet-56 (CIFAR-10)/-101 (ImageNet) to remain about 50/30/50% weights[2] without accuracy degradation, which beats the human-tuned pruning pipelines such as FPGM (He et al. (2018b)) and RL-MCTS (Wang & Li (2022)). This reduction on weights also brings about faster inference. On CIFAR-10, we can achieve maximal $3\times$ acceleration of ResNet-50 with the same accuracy, compared to original model. The boost of inference speed and the reduction of memory footprint make the application of high-accuracy models on edge devices feasible.

## 2    RELATED WORKS

In this section, we brief related works to justify the novelty of our work. We first classify prior works based on the pruning criterion, which include magnitude-/impact-based pruning. We then give an overview on works that utilizes distribution analysis. Finally, we justify the novelty of our method.

### 2.1    MAGNITUDE-BASED PRUNING

We refer magnitude-based pruning to the network-slimming approaches based on the importance of neural network's weights, which are measured by $L_1/L_2$-norm/absolute value of network's parameters/feature-maps/filters/layers (either locally or globally). Though the rationale behind them is intuitive, the methods can usually achieve outstanding pruning results with an easy-to-operate pipeline, which are extensible to be applied on different types of neural network (like Multi-layer Perceptron Hertz et al. (1991), Convolution Neural Network (CNN) Han et al. (2015b) and Transformer Mao et al. (2021)). For CNN, Han et al. (2015a)'s Deep Compression intrigues lots of follow-up works on this direction (e.g. Li et al. (2016); Gordon et al. (2020); Elesedy et al. (2020); Tanaka et al. (2020); Liu et al. (2021)). More recently, the Lottery Ticket Hypothesis Frankle & Carbin (2019) shares some similarities with this line of works: such a method assumes that a subnet, from a dense randomly-initialized neural network, can be trained separately; and it can achieve a similar level of the testing accuracy (under the same amount of training iterations). A substantial amount of efforts focus on extending this hypothesis furthermore (e.g. Zhou et al. (2019); Ramanujan et al. (2020); Malach et al. (2020); Pensia et al. (2020); Orseau et al. (2020); Qian & Klabjan (2021); Chijiwa et al. (2021a); da Cunha et al. (2022)).

### 2.2    IMPACT-BASED PRUNING

We refer impact-based pruning to methods for eliminating the weights while minimizing overheads on the model. The ancestors (LeCun et al. (1989); Hassibi et al. (1993)) of this series of works aim to find a criterion different from those described in magnitude-based approaches with a possibly more reasonable theoretical explanation. OBD LeCun et al. (1989) and OBS Hassibi et al. (1993)

---

[2]Since the discrepancy between weight reduction and FLOPs decrease is usually small, we only showcase FLOPs decrease in our experimental results (Section 5).

successfully craft a slimmer but fairly-simple model, using approximated Hessian matrix or its inverse. Follow-up works inherit their philosophy, and extensively apply to more complicated network architectures. Among them, Molchanov et al. (2019) proposes to utilize the first-order derivative to query the importance of weights, and, if needed, coarse-estimated second-order derivative can assist the decision making. After that, Singh & Alistarh (2020) leverages empirical fisher matrix to better approximate the Hessian matrix and achieve smaller accuracy degradation. Wang et al. (2020) views the problem from the perspective of gradient flow, rather than traditional weights-and-loss relation. Other works attempt to bypass the constrains of measuring weights' importance using first-/second-derivative. For example, these works score weights from the scaling factor of batch normalization layers (Liu et al. (2017)) or channel's output (Huang & Wang (2017)), and etc. A substantial amount of works utilize such a philosophy to improve the pruning performance (e.g. Lebedev & Lempitsky (2015); Molchanov et al. (2016); Yu et al. (2017); Dong et al. (2017a); Zeng & Urtasun (2018); Baykal et al. (2018); Lee et al. (2018); Wang et al. (2019); Liebenwein et al. (2019); Xing et al. (2020)).

### 2.3 Distribution-based Pruning

We define Distribution-based Pruning as the pruning pipelines diminishing the redundancy within neural network according to the distribution of a certain property (for instance, it can be the value/norm of weights, the geometric median of filters or the importance scores calculated from the second derivative of loss with respect to weights, etc) in a specific space (e.g., for scalar metrics like importance score, we can project them into a 1D number axis; for vector objects such as geometric median, a spatial n-dimensional coordinate might be more suitable). To the best of our knowledge, there are few works aware of/dedicated to this line of research. We just describe some of them obeying our definition here. Xu et al. (2020) argues that the $L_1$-norm of filters in a CONV module should obeys Gaussian Distribution, and they cut off the filters with norms distant from distribution's center. Zheng et al. (2019) and Labach & Valaee (2020) fast narrow the neural network search space according to their assumptions about networks' connection/architecture distribution. Yoshida et al. (2018) adds group LASSO into the training process to drive the network to follow their expected distribution. A statistically principled scoring strategy is proposed by Xing et al. (2020) for accurately quantifying association between networks' connections and final output. Chang et al. (2022) studies the distribution of scaling factor of BN layers in Bayesian DNN for a pruning without any other information.[3]

### 2.4 The Novelty of Our Method

The novelty of our work is two-folded. First, our work focuses on the *pruning metric search problem*, while prior arts focus on building pruning frameworks and leave the pruning metric search as hands-tuned issues. Second, our work delivers a new formalization to guide the determination of the *proper pruning metric*, which introduces new observation point called *Fabulous Coordinate*. Third, our work builds a complete solution upon our formalization, which has been quantitatively demonstrated for the effectiveness of our method.

## 3 Key Properties of Fabulous Distribution

Enlightened from Liu et al. (2018), we find that there is a meaningful weight distribution (denoted as Fabulous Distribution) hidden in a simple magnitude-based pruning process (Han et al. (2015b)). In this section, we will introduce three key characteristics of the Fabulous Distribution, which are Centralized Distribution, Retraining Recovers Centralization and Central Collapse. We also report preliminary results of our Fabulous Coordinate, the Scaled Filter Norm (SFN), which fit the aforementioned characteristics. The details of how to find this coordinate are introduced in Section 4.

---

[3]Other works also involve distribution analysis into their pipeline, but the distribution itself is not the protagonist. For instance, Ramanujan et al. (2020) tests the influence of different weight initialization schemes to their Strong Lottery Ticket Hypothesis verification; Qian & Klabjan (2021) presupposes some restrictions to parameter distribution for the convenience of their theory proof. Therefore, we consider these works don't belong to distribution-based pruning.

### 3.1 CENTRALIZED DISTRIBUTION

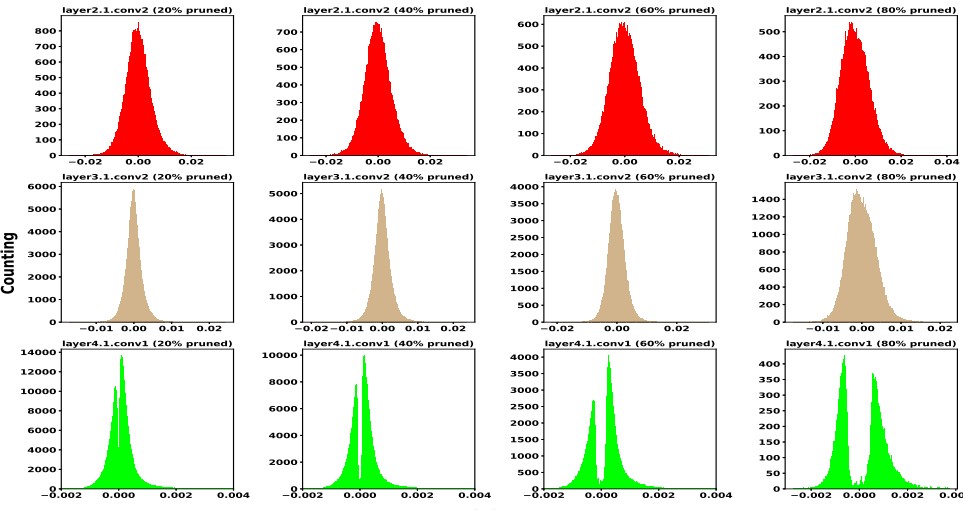

Figure 1: Centralized weight distribution for CONV layers during the prune-retrain-prune process: We select three CONV from the shallow/middle/deep layers of ResNet-50 to observe the variance of weight distribution during the pruning. Pruning pipeline and experimental settings are the same as what's described in Han et al. (2015b)'s work.

**Observation:** Figure 1 reports the distribution of weights in the form Laplace Distribution $\mathcal{L}(\mu, b)$ for each CONV layers during the prune-retrain-prune process (except the tail end of pruning when central collapse happens, which we discuss in Section 3.3). We name this phenomenon the centralization of statistics and view it as an important symbol of the overparameterized neural network. As discussed in Liu et al. (2018) and Zhong et al. (2020), this dense-in-center distribution indicates the hidden redundancy in neural networks and the objects (weights as in Liu et al. (2018), neuron as in Zhong et al. (2020)) located in the center might be able to represent each others.

**Formal Description:** For a formal description of this phenomenon, we need to first measure the degree of centralization quantitatively. Therefore, we introduce the metric, *Loose-KL-Centralization-Degree (LKL-CD)*, named after Kullback–Leibler divergence, for an accurate measure of statistics' density at the center of distribution but not restricting the distribution to be Laplace-Distribution-like. LKL-CD of distribution $P(x)$ can be calculated from the below formula:

$$LKL - CD(P(x)) = min_{(\alpha,\beta)}\Big( \int_{-\infty}^{+\infty} p(x)\, log\big(\frac{p(x)}{q_{(\alpha,\beta)}(x)}\big)\, \mathrm{d}x \Big)\ (\alpha > 0,\ \beta > 0)$$

where,

$$q_{(\alpha,\beta)}(x) = \begin{cases} \frac{\alpha}{x+\beta} & \text{if } x > 0 \\ -\frac{\alpha}{x-\beta} & \text{if } x < 0 \end{cases}$$

Then, the description of centralized distribution can be formally written as:

*We call a distribution "centralized distribution", if it's generated from a coordinate $\Psi$ and has the following property:*

$$LKL - CD(\Psi(W))\ \rightarrow\ 0$$

We quantitatively measure the centralization-degree of three representative coordinates ($L_1$-norm (Li et al. (2016)), geometric median (GM) (He et al. (2018b)) and GraSP [4] (Wang et al. (2020))), but

---

[4]GraSP is originally designed for the importance evaluation of each weight. We get the importance score of a certain filter under GraSP by gathering the per-weight scores using $L_1$-norm.

find none of them receive a satisfactory LKL-CD score. As shown in the Table 1, $L_1$-norm gains the highest score under our LKL-CD measurement. However, there is still a gap between its score and the score of Fabulous Distribution. In Figure 2, the distribution calculated by measuring $L_1$-norm appears to have multiple centers which greatly differs from our expectation. Therefore, all these three coordinates don't fit our requirement and a novel coordinate is required.

| Arch/LKL-CD | Raw Weight | $L_1$-norm | GM | GraSP | Ours |
|:---:|:---:|:---:|:---:|:---:|:---:|
| Layer1 | 0.4 | 2.3 | 3.4 | 2.5 | **1.7** |
| Layer2 | 0.7 | 1.7 | 4.0 | 1.6 | **1.0** |
| Layer3 | 0.3 | 2.1 | 2.9 | 1.7 | **1.0** |
| Layer4 | 0.5 | 2.4 | 2.8 | 2.8 | **0.9** |
| Avg. | 0.475 | 2.125 | 3.275 | 2.15 | **1.15** |

Table 1: Centralization degree comparison: We measure the LKL-CD values of each layer in a ResNet-50 model with pre-trained weights from torchvision (PyTorch).

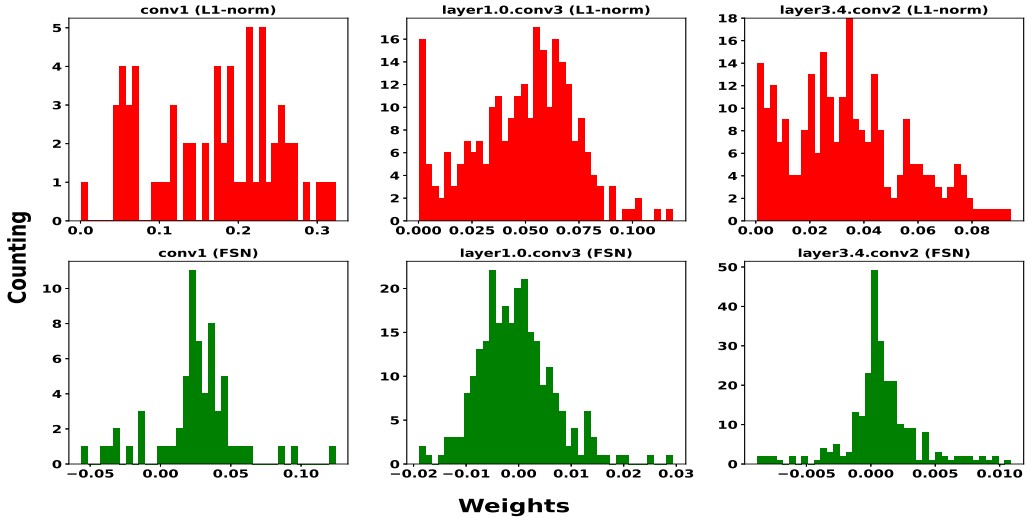

Figure 2: Distribution comparison: We compare the distributions measured by $L_1$-norm (upper) and SFN (lower) on the same model used in Table 1.

**Our Coordinate:** To address the aforementioned issue, a new coordinate, *Scaled Filter Norm (SFN)*, is found using Crossword Puzzle search engine, to get the centralized distribution. We express it in Equation 1:

$$SFN(F_W) = \mu \frac{\|F_W\|_1}{\|C_W\|_1} + \sigma \tag{1}$$

where, $F_W$ is the weight of a certain filter in CONV layer with weight $C_W$; $\mu$ and $\sigma$ are the scaling factor and bias applied to that filter in the batch normalization layer.

As shown in Table 1 and Figure 2, it not only outperforms three other coordinates in LKL-CD comparison, but also are consistent through all CONV layers. After the coordinate transfer, previous multi-center distribution becomes single-center, which demonstrates the high central density we expect. Therefore, we choose SFN as the importance scoring mechanism in our pruning pipeline.

### 3.2 RETRAINING RECOVERS CENTRALIZED DISTRIBUTION

**Observation:** The empty space created by pruning (through cutting off the central parts in the distribution) can be re-filled by retraining, which is similar to regather data at the center. Figure 3 demonstrates this phenomenon in one retraining step of pruned ResNet-50 on CIFAR-10. From Figure 3, the remaining weights distributed in the two sides migrate to the central gradually and forms a new summit in the frequency histogram. Another interesting observation is that the new-

created summit has a lower height than that of the summit before pruning. This might be explained as the consequence of redundancy decrease caused by pruning replaceable weights.

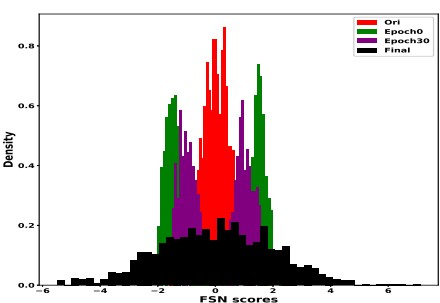

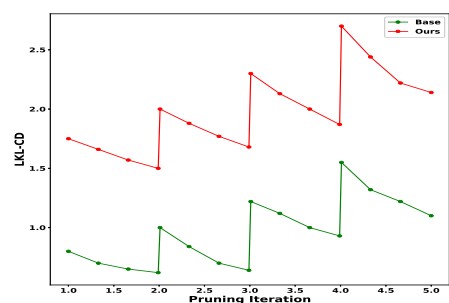

Figure 3: Distribution recovery in one retraining step: We prune pre-trained ResNet-50 and then retrain it on CIFAR-10; Epoch0 is the initial state of the retraining; Epoch30 means that we've trained the pruned model for 30 epochs.

Figure 4: Distribution recovery in several prune-retrain-prune steps: This graph depicts the LKL-CD variance during four prune-retrain-prune iterations; The per-iteration pruning rate is set to 0.2, and the retraining process goes through 40 epochs on CIFAR-10. "Base" refers to Han et al. (2015b)'s method.

**Our Coordinate:** As can be seen in the Figure 4, our prune-retrain-prune loop also experiences a similar centralization recovery process. However, compared to Han et al. (2015b)'s work, the degradation of the centralization is more apparent for our pruning. One possible explanation is that the discrepancy between our and Han et al. (2015b)'s observation coordinates prevent us to obtain an exact same distribution, and this gap is enlarged during the prune-retrain-prune process.

### 3.3 CENTRAL COLLAPSE

**Observation:** At a certain stage of prune-retrain-prune loop, retraining is unable to fill the empty space in distribution, and the original center will be filled with void (we call this phenomenon "Central Collapse"), accompanying by dramatic accuracy decline. We also find that the central collapse exists in the weight distribution of the final pruning output if we follow Deep Compression's pruning pipeline Han et al. (2015a) which is the same as the experimental results described in Han et al. (2015b)'s and Liu et al. (2018)'s work. One additional discovery is that there is a strong association between central collapse and accuracy degradation. In Figure 5, central collapse and the turning point of accuracy-pruning-rate-line nearly occur at the same time. We believe this is not a coincidence, since the central collapse possibly also indicates close-to-zero redundancy in the current layer. Therefore, further pruning on this "already succinct" layer will greatly hurt the accuracy.

**Our Coordinate:** We also encounter the central collapse and the experimental evidence are depicted in Figure 5. After pruning 50% weights out of the model, the central collapse becomes noticeable in some layers and our model accuracy goes down simultaneously. For a closer observation, we provide a break-down FSN distribution of representative CONV layers in Figure 6. From Figure 6, most layers suffer irreversible damages and fall into a distribution with central collapse.

## 4 CROSSWORD PUZZLE SEARCH ENGINE

In this section, we introduce the pipeline of our Crossword Puzzle search engine, which takes a pre-trained neural network and pruning metric building block as inputs and outputs a Fabulous Coordinate.

**Neighbor Coordinate Generator:** First, we generate a coordinate according to the input pruning metrics, by taking the linear combination of several intuitively selected criteria. For instance, this generator might produce a coordinate, $\Psi(F_W) = \|F_W\|_1 + \|F_W\|_2$, where $F_W$ is the weight of filters. In practice, we classify these building blocks according to magnitude-/impact-/distribution-based pruning and begin searching within these clusters first, as metrics derived from methods in

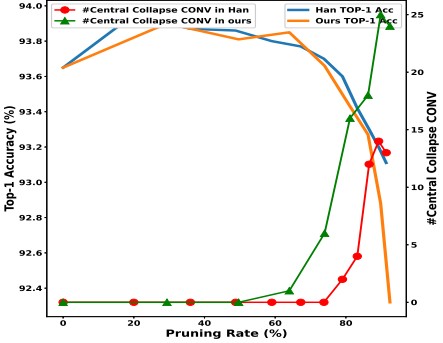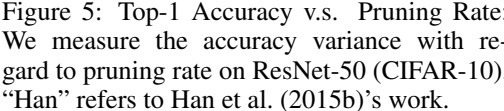

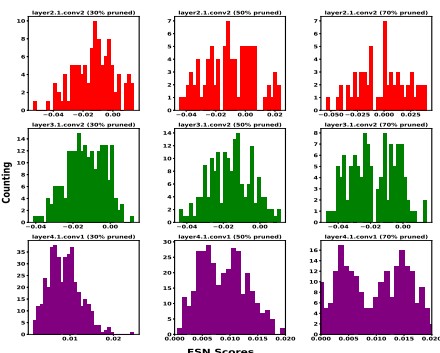

Figure 5: Top-1 Accuracy v.s. Pruning Rate: We measure the accuracy variance with regard to pruning rate on ResNet-50 (CIFAR-10). "Han" refers to Han et al. (2015b)'s work.

Figure 6: The FSN distribution of representative CONV layers of Resnet-50 (Pruned and retrained on CIFAR-10). Similar characteristics are observed on other layers.

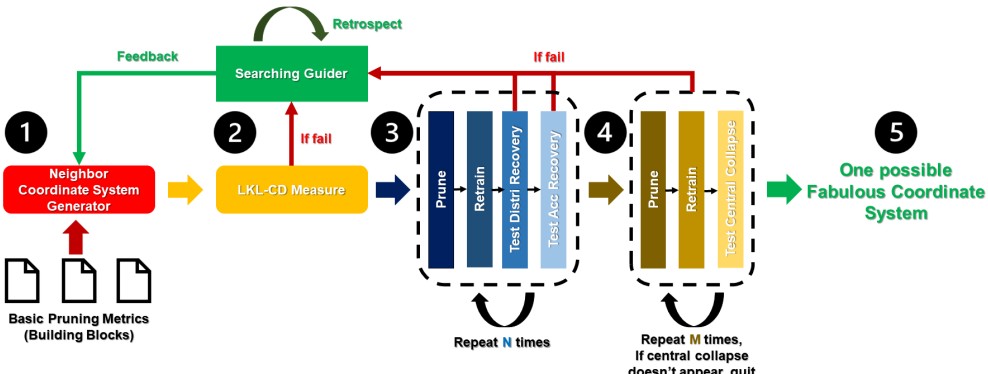

Figure 7: Crossword Puzzle Search Engine: Input: 1) building blocks of pruning metrics (e.g., the $L_1$-norm of filter's weights) and 2) the pre-trained neural network (not shown in this figure); Output: a pruning metric on which Fabulous Distribution can be observed.

a same cluster are expected to have strong association. Note that during searching, our crossword puzzle engine don't need to strictly obey this intuitive rule since it can jump out of it when triggered by the simulated annealing search guider.

**Searching Guider:** We add a simulated annealing (Kirkpatrick et al. (1983)) search guider to steer the neighbor coordinate generator. The possibility of accepting a coordinate is expressed as $p = e^{-v/t}$, where $t$ is the temperature and $v$ is the LKL-CD value. From the equation, we can see that, a coordinate with high LKL-CD value still can be accepted as the start point of the next neighbor search. In practice, this annealing approach enables the engine to expand search space and find hybrid solutions.

**Centralization Degree Filter:** We set a filter to halt coordinates with high LKL-CD values from continued searching. According to the coordinate, the filter can calculate the average LKL-CD value of the whole model. If the LKL-CD value exceeds the threshold $A$ (*a preset constant*), we can drop this coordinate and inform the searching guider, since the high LKL-CD value directly violates our definition for Fabulous Coordinate. Experimental results show that this filter can filter out more than 80% coordinates, speeding up the searching process greatly.

**Retraining Recovery Verification:** The second stage is to conduct some prune-retrain-prune trials for the verification of the retraining recovery ability of a certain coordinate. In practice, we use the following equation to measure this ability:

$$Recovery\_Ability = A \times Avg(LKL - CD(\Psi(W))) + B \times \Delta Acc\_TOP\_1$$

where, $Avg(LKL - CD(\Psi(W)))$ is the average LKL-CD value of 10 prune-train-prune iterations, each with 20 epochs' retraining; $\Delta Acc\_TOP\_1$ is the top-1 accuracy improvement during this procedure; $A$ and $B$ are two scaling constants.

**Central Collapse Inspector:** Finally, we test the existence of central collapse and output this coordinate as a possible Fabulous Coordinate if it passes all the tests. To track the occurrence of central collapse, we closely monitor the variance of LKL-CD during the prune-retrain-prune process. The central collapse happens when an increment of more than 1.0 can be observed on LKL-CD,.

## 5 EXPERIMENTAL RESULTS

We evaluate our pruning method with SFN coordinate for two popular convolutional neural networks VGG-16-bn (Simonyan & Zisserman (2014)) and ResNet-50/-56/-101/-110 (He et al. (2015)) tested on CIFAR-10 (Krizhevsky (2009)) and ImageNet (Deng et al. (2009)) with details of experimental settings and results displayed below.

### 5.1 EXPERIMENT SETTINGS

In this section, we elaborate the pruning and retraining settings. For pruning, we utilize PyTorch (Paszke et al. (2019)) and one third-party library Torch Pruning (Fang (2022)). At the beginning of each prune-retrain-prune iteration, we cut off 10% filters in each CONV layer, which results in about 30% weights being pruned off in the whole model, due to the connection between CONV layers (more details in Section A). For retraining, we inherit the same training settings and functions from two standard examples (Phan (2022); PyTorch (2022)), except that the training epoch is adjusted to 100/120 on CIFAR-10/ImageNet to achieve a fair competition under the same computation budget (as suggested by Liu et al. (2018)). All experiments are conducted on a machine with NVIDIA RTX A5000 and Intel Xeon Gold 6330.

### 5.2 RESULTS ON CIFAR-10

Our auto-generated pruning pipeline can achieve comparable pruning results on CIFAR-10 with regard to other human-built methods (such as PFEC (Li et al. (2016)), MIL (Dong et al. (2017b)), CP (He et al. (2017)), SFP (He et al. (2018a)), FPGM (He et al. (2018b)), PFGDF (Xu et al. (2020)), FPPMO (Li et al. (2021)), RL-MCTS (Wang & Li (2022))).

As shown in Table 2, our method can reduce the FLOPs of ResNet-56/110 by 73.0%/63.0% with slightly accuracy improvement, while, on VGG-16-bn, we can't defeat the state-of-the-art human-created pruning frameworks, PFGDF and RL-MCTS. The success on ResNet might be due to the specific pruning strategy (in Section A) we apply to shortcut connections. This strategy takes the CONV layers associated by shortcut connections as a whole, and, therefore, reveals useful global information to our pruning, which assists us to prune more accurately. However, VGG-16-bn doesn't possess such structure, and the limited information impair the effect of our pruning (by default, our pruning is performed solely on each CONV layer). This might demonstrates that our proposed pruning pipeline (in Section A) can be further enhanced by considering the global environment. But the focus of this work is not on any dedicated pipeline, so we just use a simple one.

### 5.3 RESULTS ON IMAGENET

On ImageNet, our pruning framework can also compete with methods fine-tuned by human (for instance, ThiNet (Luo et al. (2017)), SFP (He et al. (2018a)), Rethinking (Ye et al. (2018)), FPGM (He et al. (2018b)), FPPMO (Li et al. (2021)), RL-MCTS (Wang & Li (2022))).

Table 3 demonstrates that our method can outperform the state-of-the-art methods with more accuracy being preserved, given similar FLOPs drop. We owe this positive outcome to the strength of Fabulous Distribution. When the data amount goes up, the Fabulous Distribution can reveal the redundancy more accurately, since the hidden characteristics within statistics are more likely to be discovered when there are numerous samples.

| Model | Method | Baseline Top-1 Acc (%) | Top-1 Acc after Pruning (%) | △Top-1 Acc (%) | △FLOPs (%) |
|-------|--------|-----|-----|-----|-----|
| ResNet-56 | PFEC | 93.04 | 93.06 | +0.02 | -27.6 |
| | CP | 92.80 | 91.80 | -1.00 | -50.0 |
| | SFP (w/o pre-trained) | 93.59 | 92.26 | -1.33 | -52.6 |
| | FPGM (GM-only 40%) | 93.59 | 93.49 | -0.10 | -52.6 |
| | FPPMO (45%) | 93.59 | 93.50 | -0.09 | -58.0 |
| | RL-MCTS | 93.20 | 93.56 | **+0.36** | -55.0 |
| | **Ours (SNF)** | 93.65 | 93.9 | **+0.25** | -28.6 |
| | | | 93.66 | **+0.01** | **-73.0** |
| ResNet-110 | MIL (w/o pre-trained) | 93.63 | 93.44 | -0.19 | -34.2 |
| | PFEC | 93.53 | 93.30 | -0.23 | -38.6 |
| | SFP (w/o pre-trained) | 93.68 | 93.38 | -0.30 | -40.8 |
| | FPGM (mix 40%) | 93.68 | 93.85 | **+0.17** | -52.3 |
| | FPPMO (45%) | 93.68 | 93.76 | +0.08 | -57.7 |
| | **Ours (SNF)** | 93.68 | 93.97 | **+0.29** | -48.7 |
| | | | 93.74 | +0.06 | **-63.0** |
| VGG-16-bn | PFEC | 93.58 | 93.31 | -0.27 | -34.2 |
| | FPGM | 93.58 | 93.54 | -0.04 | -34.2 |
| | PFGDF | 93.25 | 93.48 | **+0.23** | **-70.27** |
| | RL-MCTS | 93.51 | 93.90 | **+0.39** | -45.5 |
| | **Ours (SNF)** | 94.00 | **94.29** | +0.29 | -53.1 |

Table 2: Pruning results on CIFAR-10: "pre-trained" means whether to use pre-trained model as the initialization or not. Other labels like "GM-only" keep the same meanings as those in original papers. Our competitors' data are collected from their released codes/model/best results on paper.

| Model | Method | Baseline Top-1/5 Acc (%) | Top-1/5 Acc after Pruning (%) | △Top-1/5 Acc (%) | △FLOPs (%) |
|-------|--------|-----|-----|-----|-----|
| ResNet-50 | ThiNet | 72.88/91.14 | 72.04/90.67 | -0.84/-0.47 | -36.7 |
| | SFP | 76.15/92.87 | 62.14/84.60 | -14.01/-8.27 | -41.8 |
| | FPGM (GM-only 30%) | 76.15/92.87 | 75.59/92.63 | -0.56/-0.24 | -42.2 |
| | FPPMO (40%) | 76.15/92.87 | 74.91/92.39 | -1.24/-0.48 | **-53.5** |
| | RL-MCTS | 77.34/93.27 | **76.80/93.00** | -0.54/-0.27 | -46.1 |
| | **Ours (SNF)** | 76.13/92.86 | 75.65/92.64 | **-0.48/-0.22** | **-49.3** |
| ResNet-101 | Rethinking | 77.37/93.56 | 75.27/- | -2.10/- | -47.0 |
| | FPGM (GM-only 30%) | 77.37/93.56 | 77.32/93.56 | -0.05/0.00 | -42.2 |
| | FPPMO (35%) | 77.37/93.56 | 77.18/93.52 | -0.05/0.00 | **-50.2** |
| | **Ours (SNF)** | 77.37/93.55 | **77.54/93.68** | **+0.17/+0.13** | -49.5 |

Table 3: Pruning results on ImageNet: labels have the same meanings as those in Table 2. Our competitors' data are collected from their released codes/model/best results on paper.

## 6 CONCLUSIONS

We propose The Cross Puzzle, a new method to simplify this procedure by automatically deriving pruning metrics. The key insight behind our method is that: *For Deep Neural Network Models, a Pruning-friendly Distribution of model's weights can be obtained, given a proper Coordinate.* We experimentally confirm the above insight, and denote the new Coordinate as the Fabulous Coordinates. Our quantitative evaluation results show that: the Crossword Puzzle can find a simple yet effective metric, which outperforms the state-of-the-art pruning methods by delivering no accuracy degradation on ResNet-56 (CIFAR-10)/-101 (ImageNet), while the pruning rate is raised to 70%/50% for the respective models.

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

# A APPENDIX: THE PROPOSED PRUNING PIPELINE

Like many previous practices (Han et al. (2015a;b); Li et al. (2016); Frankle & Carbin (2019); Chijiwa et al. (2021b)), we also prune the neural network iteratively (as shown in Algorithm 1). However, the pruning step is slightly different from that of magnitude-/impact-based pruning frameworks, since we prune redundant parts according to distribution analysis while handling shortcut connections specially.

---

**Algorithm 1** Iterative Pruning

---

**Input:** pre-trained model with weights $W$; pruning rate $PR$; prune times $PT$; retrain epochs $Ep$
**Output:** pruned model weights $W$

1: **function** ITERATIVE_PRUNING($W, PR, PT, Ep$)
2:     **for** $i \leftarrow 1$ to $PT$ **do**
3:         $D \leftarrow Analyse\_Distri(W)$
4:         $M \leftarrow Prune\_Center(D, PR)$                                     ▷ $M$ is the mask
5:         $W \leftarrow W \odot M$                           ▷ prune model according to mask
6:         $Train(W, Ep)$                            ▷ retrain model for $Ep$ epochs
7:     **end for**
8:     **return** $W$
9: **end function**

---

**Pruning According to Distribution:** We clip off the central parts in the distribution, which can be interpreted as the generalization of traditional magnitude-based pruning. Some classic magnitude-based pruning practices (such as Han et al. (2015a); Liu et al. (2017)) hold the belief, smaller-value-less-important. However, we owe the success of these works to the coincidence that small values happen to lie in the center of distribution. For instance, in Figure 1, the center of distribution overlaps the original point in the horizontal axis. The pruning of small values also takes the effect of cutting down the peak of distribution. Based on these observations, we claim that what's matter most is the center of distribution rather than the small values. Some experiments are conducted to verify this conjecture. As can be seen in Figure 8, pruning random/smallest/largest/two-sides values in the distribution doesn't result in a better outcome than pruning the center parts. This might point to the fact that the center of distribution has a special meaning.

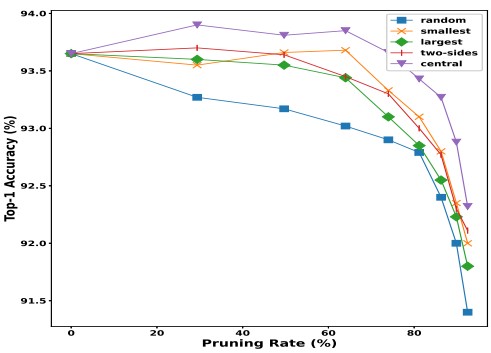
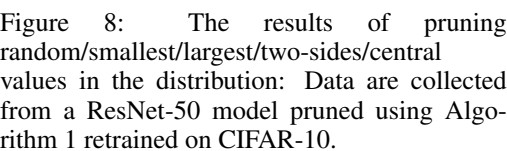
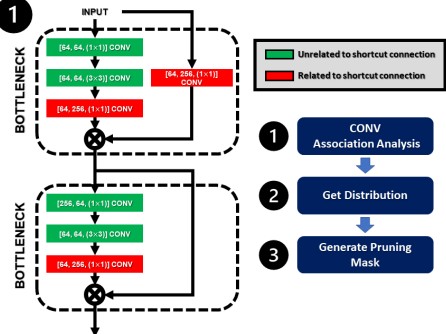

Figure 8: The results of pruning random/smallest/largest/two-sides/central values in the distribution: Data are collected from a ResNet-50 model pruned using Algorithm 1 retrained on CIFAR-10.

Figure 9: Prune CONV related to shortcut connections in ResNet-50: Last CONV layer of the bottleneck module and the downsample CONV layer are associated together through the shortcut connections. When pruning one of them, the others need to be adjusted accordingly.

**How to Handle Shortcut Connection:** Our pruning framework can prune the CONV layers around shortcut connections, an essential component in many advanced CNN models like ResNet He et al. (2015) and DenseNet Huang et al. (2016), by taking the computation association into consideration.

In Figure 9, we illustrate how to prune the CONV layers in ResNet. First, we analyse the dependency of CONV layers to pick out the ones which should be pruned together. Then, by treating these layers as a whole, we can get the distribution and corresponding pruning mask. Finally, the mask is applied to these layers and the pruning can proceed to other layers unrelated to shortcut connections.

