# OpenReview forum: "The Crossword Puzzle: Simplifying Deep Neural Network Pruning with Fabulous Coordinates"
_ICLR.cc/2023/Conference — Submitted to ICLR 2023_

### Official Review · Reviewer_bFZJ · 2022-10-24

**Confidence:** 3
**Correctness:** 3
**Technical Novelty And Significance:** 3
**Empirical Novelty And Significance:** 3
**Recommendation:** 6

**Clarity, Quality, Novelty And Reproducibility:**

The paper needs clarifications in a number of places.

•	How should one understand the second requirements of the Fabulous Coordinates, that retraining recovers centralized distribution? This seems to contradict to the third requirement (central collapse). In the Figure 7 pipeline, a coordinate goes through retraining recovery verification first, then to central collapse inspector. These two procedures seem to do similar things, i.e., pruning, retraining, and then test distribution. How can a coordinate gives recovered center in the first procedure, while gives collapsed center in the second procedure?

•	 In Section 3.1, the paper proposes the LKL-CD measurements. Table 1 highlights the value of it. Why does a lower value indicate a better measurement?

•	In Figure 4, how can one conclude that the degradation of centralization is more apparent for the proposed pruning method? From the figure, the LKL-CD value of the method is higher than the base method.

•	There is no information on how the pruning metric blocks are build. For example, in Section 4, the search engine can jump out of the given intuitive rule when trigged by the search guider, while it is unclear how the search guider behaves. The only description for it is the acceptance probability.

•	Following the previous question, the description for other parts of the pipeline is also ambiguous. It is strongly suggested that the paper should include more details of them. The implementation code is not included, which makes the reviewer harder to understand each part.


**Strength And Weaknesses:**

Strengths

•	The paper gives a comprehensive introduction to previous work on pruning metrics categorized in 3 different groups, magnitude, impact, and distribution-based methods. These build the metrics blocks for search.
•	The paper introduces the Fabulous Coordinate with three properties, a new perspective of understanding the model redundancies.

Weaknesses

•	There is little explanation on why the proposed LKL-CD is better than other metrics. The paper attempts to explain it through Table 1, the four layers in ResNet-50. It might not be statistically enough to claim the advantage of it using one type of network. More validation should be done either empirically or theoretically. It would be better to see that comparisons between them are included in the experiment section.


**Summary Of The Paper:**

This paper proposes a Crossword Puzzle engine for searching the optimal pruning metrics automatically. It first gives three properties of a good metric/coordinate: centralized distribution,  retraining recovers centralized distribution and central collapse. Then, it introduces a metrics called LKL-CD  for measuring the centralization degree. The paper finally presents their coordinate obtained from the engine and empirically evaluates the effectiveness of the coordinate system for pruning.

**Summary Of The Review:**

The paper proposes novel perspective of pruning. Intensive experiment is done to search better coordinate systems to guide the pruning procedure. However, some key points need further and clearer explanation.

---

> ### Author Response · Authors · 2022-11-12
> **Response to Reviewer bFZJ**
>
> We thank reviewer bFZJ for his/her reasonable criticism and suggestions.
>
> Q1: There is little explanation on why the proposed LKL-CD is better than other metrics.
>
> A1: LKL-CD is a neutral criterion. The other metrics like the Kullback–Leibler divergence between one distribution (X) and Gaussian/Laplace distribution are not suitable to measure the centralization degree of X since they might make wrong assumption of X’s distribution pattern. Using them will implicitly assume that X should be similar/equal to Gaussian/Laplace distribution, which will impose unreasonable premises on our experiments. Therefore, we choose to use the neutral criterion, LKL-CD, to measure the centralization degree without any pre-assumption about the distribution pattern. We will use experimental results to demonstrate this point in the next version.
>
> Q2: How should one understand the second requirements of the Fabulous Coordinates, that retraining recovers centralized distribution? This seems to contradict to the third requirement (central collapse).
>
> A2: The “retraining recovers centralized distribution” is the distribution characteristic observed during our prune-retrain-prune loop, and this recursive pruning process ends in “central collapse”. When there are considerable redundancy within the model, we can observe the “retraining recovers centralized distribution” during retraining. But when we repeatedly prune filters out and the redundancy decrease to a certain level, the “central collapse” will happen, and the model can’t recover the centralized distribution through retraining. “central collapse” is a signal that the model might have already been succinct and can’t be further pruned. Once the “central collapse” happens, we will stop pruning.
>
> Q3: In Section 3.1, the paper proposes the LKL-CD measurements. Table 1 highlights the value of it. Why does a lower value indicate a better measurement?
>
> A3: LKL-CD is adapted from Kullback–Leibler divergence. Kullback–Leibler divergence is a common metric for measuring the similarity between two distributions P and Q (denoted $D_{KL}(P\|Q)$). When $P=Q$, $ D_{KL}(P\|Q)=0$. In this paper, we expect the Fabulous Distribution ($P$) to be a centralized distribution. Therefore, we quantify the similarity between Fabulous Distribution ($P$) and a centralized distribution ($Q$, where Q’s definition is provided in Section 3.1) using LKL-CD.
>
> Q4: In Figure 4, how can one conclude that the degradation of centralization is more apparent for the proposed pruning method? From the figure, the LKL-CD value of the method is higher than the base method.
>
> A4: Compared to our method, the distribution of base method (Han et al., 2015) is closer to our definition of Fabulous Distribution, and this leads them to get lower value in LKL-CD measurement. This paper observes and defines the Fabulous Distribution based on Han et al., 2015. Han et al., 2015 have an advantage in getting a lower LKL-CD value by its nature. Our work aims to find another Fabulous Coordinate which differs from Han et al., 2015, but can’t promise the found Fabulous Coordinate is more “fabulous” than Han et al., 2015.
>
> Q5: There is no information on how the pruning metric blocks are build and how to jump out of the block.
>
> A5: For the components of pruning metric blocks, we have provided a simple explanation in Section 4. A more detailed explanation will be added in future. For the working mechanism of search guider, it guides the searching to jump to another block through probability. For instance, we have already searched 80% combinations in "magnitude-based pruning criteria block", and then, for the next iteration, the guider will ask the neighbor coordinate generator to involve new criterion from another block by a chance of 80% (we use the simple number 80% here just for the convenience of explanation). We will add more details of this in future.
>
> Q6: Following the previous question, the description for other parts of the pipeline is also ambiguous.
>
> A6: We find our paper lacks the necessary description of the proposed search engine. We will add an appendix/section to reveal more details in the next version. Thanks for your suggestions.
>
> Q7: The implementation code is not included, which makes the reviewer harder to understand each part.
>
> A7: We plan to release our code in the next version.

---

### Official Review · Reviewer_TNSn · 2022-10-25

**Confidence:** 4
**Correctness:** 3
**Technical Novelty And Significance:** 3
**Empirical Novelty And Significance:** 2
**Recommendation:** 5

**Clarity, Quality, Novelty And Reproducibility:**

Clarity
The explanation of the method and important observations are good. However, certain portions of the measures and evaluation scheme is slightly obscure.

Quality and Novelty
The measure and overall search framework offer a requisite level of novelty and quality to the proposed work. However, a deeper evaluation is necessary to highlight these ideas.

**Strength And Weaknesses:**

Strengths
- The proposed work offers an interesting idea on measuring the quality of a proposed pruning metric.
- Combined with an almost meta-learning-like framework on identifying a high quality pruning metric, automation of this domain is certainly important and brings pruning a step closer to being part of the development cycle of DNNs.

Weaknesses
- Could the authors provide more quantitative bounds when mentioning "relatively-high pruning rate" in Pg. 1, footnote.
- When highlighting the improvement in acceleration, could the authors clarify whether the "3x" improvement is in theoretical FLOPs or real-time inference latency?
- I would encourage the authors to look closely into the literature discussing distribution-based pruning. Apart from the papers mentioned in the current manuscript, there are broad swathes of work on probabilistic pruning that can and should be referenced.
- Could the authors provide an intuition for the formulation of the LKL-CD, including how they solve for the final values?
- Across multiple instances there are mix ups between the terminology of SFN and FSN (E.g., Fig. 2, Fig. 3 and others). Please correct them.
- The LKL-CD of Raw Weights in Table 1 provide a much better approximation than the chosen SFN formulation, assuming lower values are better. Could the authors justify the selection of SFN?
- Could the authors clarify the meaning of the notation $C_W$?
- I encourage the authors to increase the font of legends and Axes ticks for readability.
- Quantifiably, could the authors provide metrics used to measure the collapse as well as highlight how exactly they were applied to measure the values provided in Fig. 5?
- In addition, could the authors provide more comparisons on the recovery process of the central hole across the various conv layers?
- From Tables 2 and 3,
     - Across VGG16 and Table 3, SFN provides better accuracy at a lower % of FLOPs removed when compared to previous approaches.
     - Could the authors clarify their choice in prior art and comparisons? Since there are a number of previous works with higher performances that haven't been included.
     - In addition, could the authors include the decrease in number of parameters as well, since it provides a slightly different perspective on the degree to which a network can be pruned.


**Summary Of The Paper:**

Existing pruning approaches rely on intuition- or experience-based measures to determine the constraint based on which to prune neural networks. Instead, the current work proposes a framework, Crossword Puzzle, to guide the search of a pruning criterion given basic building blocks and evaluate the quality of the pruning criterion when applied on networks. Specifically, the proposed framework is based on the Fabulous coordinates, which indicate that when the distribution from a provided model match the constraints set for these coordinates, they are highly qualified to provide good pruning outcomes.

**Summary Of The Review:**

While the core ideas are relatively straightforward, the evaluation and explanation of certain aspects of the work are obscure. Addressing the weaknesses mentioned above will bring more clarity to the results and content of the proposed work.

---

> ### Author Response · Authors · 2022-11-12
> **Response to Reviewer TNSn**
>
> We thank reviewer TNSn for his/her reasonable criticism and suggestions.
>
> Q1: Could the authors provide more quantitative bounds when mentioning "relatively-high pruning rate" in Pg. 1, footnote.
>
> A1: Sorry for the ambiguous term "relatively-high pruning rate" used here. We will use more accurate terms in the next version. The quantity of "relatively-high pruning rate" will change according to the models/datasets we use. For instance, for ResNet-50 (ImageNet), the state-of-the-art baseline is the RL-MCTS (2022) which achieves about 45% pruning rate with negligible accuracy degradation (-0.54%). But for ResNet-56 (CIFAR-10), the standard is created by FPPMO (2021). It verifies that about 60% weights can be pruned out while not influencing the model accuracy.
>
> Q2: When highlighting the improvement in acceleration, could the authors clarify whether the "3x" improvement is in theoretical FLOPs or real-time inference latency?
>
> A2: It’s the improvement in real-time inference latency. In the next version, we will provide the experimental results of real-time acceleration.
>
> Q3: I would encourage the authors to look closely into the literature discussing distribution-based pruning. Apart from the papers mentioned in the current manuscript, there are broad swathes of work on probabilistic pruning that can and should be referenced.
>
> A3: We will revise this section in the next version. Thanks for your thoughtful suggestions!
>
> Q4: Could the authors provide an intuition for the formulation of the LKL-CD, including how they solve for the final values?
>
> A4: We learn this formula empirically. In experiments (not shown in this paper), we also try to use Kullback–Leibler divergence of Gaussian Distribution and Laplace Distribution but find none of them can help our search engine get appropriate pruning criteria. We will show comparison of different choices in the next version. For the calculation process of LKL-CD, we also plan to add a section to explain it in the future.
>
> Q5: Across multiple instances there are mix ups between the terminology of SFN and FSN (E.g., Fig. 2, Fig. 3 and others). Please correct them.
>
> A5: We will carefully rewrite our paper.
>
> Q6: The LKL-CD of Raw Weights in Table 1 provide a much better approximation than the chosen SFN formulation, assuming lower values are better. Could the authors justify the selection of SFN?
>
> A6: Sorry for using the confusing word “Raw Weights”. The “Raw Weights” refers to the pruning method of Han et al. (2015). It directly measures the importance of each weight based on their absolute values. This simple measurement indeed gets a higher LKL-CD score and achieves promising results. For instance, in Table 1 and Figure 4/5, it outperforms our method in some cases. But please note that we prune filters while Han et al. prune weights. This ensures our method can achieve runtime inference acceleration without any support library.
>
> Q7: Could the authors clarify the meaning of the notation $C_W$?
>
> A7: $C_W$ is the weight of a certain CONV module. In that module, we have several filters ($F$) whose weights are denoted by $F_W$.
>
> Q8: I encourage the authors to increase the font of legends and Axes ticks for readability.
>
> A8: We will fix these problems in the next version.
>
> Q9: Quantifiably, could the authors provide metrics used to measure the collapse as well as highlight how exactly they were applied to measure the values provided in Fig. 5?
>
> A9: When the LKL-CD value of a certain CONV layer increases by more than 1.0, we will say the central collapse happens in that CONV layer. The “#Central Collapsed CONV” in Fig.5 is the number of CONV layers where central collapse occurs in current/previous prune-retrain-prune iterations.
>
> Q10: In addition, could the authors provide more comparisons on the recovery process of the central hole across the various conv layers?
>
> A10: We plan to add a section to discuss this in the next version.
>
> Q11: Could the authors clarify their choice in prior art and comparisons? Since there are a number of previous works with higher performances that haven't been included.
>
> A11: We compare our method to the state-of-the-art works in filter-wise pruning. Sorry for the unclear statement in our paper.
>
> Q12: In addition, could the authors include the decrease in number of parameters as well, since it provides a slightly different perspective on the degree to which a network can be pruned.
>
> A12: Thanks for your advice. We will add related experimental results in the future.

---

> > ### Comment · Reviewer_TNSn · 2022-11-17
> > **Thanks for the response**
> >
> > Dear Authors,
> > Thank you for providing a detailed response to the feedback. Given the current state of the manuscript, it requires more modifications. All the feedback provided should help improve the next version of the paper significantly.
> > Good luck.

---

### Official Review · Reviewer_jFro · 2022-10-25

**Confidence:** 4
**Correctness:** 1
**Technical Novelty And Significance:** 1
**Empirical Novelty And Significance:** Not applicable
**Recommendation:** 1

**Clarity, Quality, Novelty And Reproducibility:**

* Poor clarity, unable to read
* Not sure about novelty because it is difficult to get useful information from the paper.

**Strength And Weaknesses:**

[Weakness]

The paper is poorly written. It looks like the paper is generated by some Google translation engine with lots of meaningless words. I tried my best but honestly speaking, I cannot understnad what the authors are writing about. In Section 3, the authors throwed up lots of formulars and figures without sufficient explanation. I have no clew what these numbers or equations mean.

**Summary Of The Paper:**

The authors proposed a so-called Crossword Puzzle method to find the optimal criteria for network pruning. The key idea is to find a so-called Fabulous Coordinate which satisfies three key properties for pruning. The authors validated their method on ImageNet and show that they can compress ResNet family by 50% without accuracy degradation.

**Summary Of The Review:**

Strong reject.

---

> ### Author Response · Authors · 2022-11-12
> **Response to Reviewer jFro**
>
> We thank reviewer jFro for his/her reasonable criticism.
>
> Q1: The paper is poorly written.
>
> A1: We will address the writing issues in the next version. Sorry for the inconvenience caused to you.

---

### Official Review · Reviewer_bVnh · 2022-10-25

**Confidence:** 5
**Correctness:** 1
**Technical Novelty And Significance:** 1
**Empirical Novelty And Significance:** 1
**Recommendation:** 1

**Clarity, Quality, Novelty And Reproducibility:**

The paper is very difficult to read.

A partial list of edits:
edits:
p. 2:
"cutting off by pruning" --> "cut off by pruning"
"nearly none redundancy " --> "nearly no redundancy"
"intrigues lots of follow-up works" --> "inspired many follow-up works"
"amount of efforts" --> "amount of effort"
"We refer impact-based pruning to methods for ... ." --> "We refer to methods for ... as impact-based pruning."
"coarse-estimated second-order derivative" --> "a coarsely-estimated second-order derivative"?
p. 3:
" leverages empirical fisher matrix" --> " leverages the empirical Fisher matrix"
"constrains" --> "constraints"
", and etc" --> delete this.
"aware of/" --> delete this.
"should obeys Gaussian Distribution" --> unclear. Maybe: "should approximate a Gaussian Distribution"?
"fast narrow the neural network" --> unclear. Maybe "first narrow the neural network"?
"in Bayesian DNN " --> "in a Bayesian DNN "
"two-folded" --> "two-fold"
"as handstuned issues" --> unclear. Maybe "for hand-tuning".
"introduces new observation point" --> "introduces a new observation point"
"distribution itself is not the protagonist" --> unclear
"​​to parameter distribution" --> "to parameter distributions"?
p. 4:
"weights in the form Laplace " --> unclear. Is the suggestion that the weights are distributed approximately as a Laplace distribution?
"neuron as" --> "neurons as"?
 "represent each others" --> "represent each others"
"coordinate psi" --> Should this be "parameter psi"?
p. 5:
"don’t fit our requirement" --> it is unclear to me what the requirement is.
Figure 1 and 2: vertical axis would be better labeled "Count" than "Counting"

"Crossword Puzzle search engine" --> unclear to me what this is


**Strength And Weaknesses:**

The paper is very difficult to read. Though experiments are performed, the experiments are not up to the quality expected. For example, there is no direct comparison with The Lottery Ticket Hypothesis paper (https://arxiv.org/pdf/1803.03635.pdf), which can reduce network size by over 90%.

Terms are not clearly defined. I cannot find clear definitions of the Cross Puzzle, the Fabulous Coordinate or the Fabulous Distribution.



**Summary Of The Paper:**

The paper proposes the Cross Puzzle to find something called the Fabulous Coordinate for pruning neural networks.

**Summary Of The Review:**

Overall, this paper id not ready for publication.
The key idea is not clearly communicated, the terms are not well defined, and there are not experiments that support the key idea.

---

> ### Author Response · Authors · 2022-11-12
> **Response to Reviewer bVnh**
>
> We thank reviewer bVnh for his/her reasonable criticism.
>
> Q1: The paper is very difficult to read.
>
> A1: We will address the writing issues in the next version. Sorry for the inconvenience caused to you.
>
> Q2: There is no direct comparison with The Lottery Ticket Hypothesis paper, which can reduce network size by over 90%.
>
> A2: This paper has nothing to do with Lottery Ticket Hypothesis. First, our pruning criterion “SFN” measures the importance of filters while Lottery Ticket Hypothesis evaluates the importance of each weight parameter. Second, we focus on improving the accuracy of the pruned model rather than the effective training from the start, one main target of Lottery Ticket Hypothesis.

---

### Official Review · Reviewer_uiqw · 2022-10-27

**Confidence:** 4
**Correctness:** 2
**Technical Novelty And Significance:** 2
**Empirical Novelty And Significance:** 2
**Recommendation:** 3

**Clarity, Quality, Novelty And Reproducibility:**

Poor presentation and quality. While novelty exists in the motivation, it is executed in an unscientific fashion. It seems like a paper thrown all together at the last moment and I do not recommend publishing of the paper even after significant registrations.

**Strength And Weaknesses:**

Strength:
1) The problem paper tries to solve is real and has real-world utility.

Weakness:
1) The abstract is extremely uninformative, so is the introduction.
2) There are terms used without any explanation or pointers to -- what is a coordinate? is something it too me a long to process
3) What is fabulous about the distribution?
4) The motivation for the design principles come from unstructured pruning and the paper experiments on structured pruning, I am not sure how that works. I know the connections between both styles exist, but connecting Han et al 2015 to the current paper does not gel well.
5) I have no idea how the candidates for coordinates are even generated for search, the entire pipeline is so under-explained at required places and over-exposited at places with little impact on the entire method.
6) Table 1 is extremely hard to comprehend.
7) definitions of equations follow the same suit.
8) Figure 5 shows your method is more inaccurate than Han et al., I do not understand.
9) The experiments are performed on limited evaluations and at 50% structured sparsity which is not even considered to be sparse because of the known redundancies. The usual comparisons happen at over 70% for ImageNet.
10) FLOPS is not directly proportional to sparsity due to non-uniformity across layers, so that metric needs to be fixed.
11) While the figure are slightly accessible, they are difficult to read due to poor placement and design.

**Summary Of The Paper:**

This paper proposes a new method called  "The Cross(word) Puzzle" that helps in getting better pruning metrics to further improve pruned neural networks in accuracy for the same compute footprint. This is informed by what the authors call Fabulous Coordinates (the choice of pruning criteria) and Fabulous distribution (the target distribution of weights after pruning).

The paper starts of by motivating what the useful properties of a good pruning function should look like and leverages it to build a search pipeline for semi-automating it. They evaluate the method with a couple of experiments on ImageNet and CIFAR-10


The brevity of the review is due to the fact that the paper is extremely incoherent and it took me over 5 hrs to digest the information. The paper is extremely poorly written, with no solid definitions, inconsistent naming (SFN vs FSN vs SNF -- what is this?), poor naming (Fabulous?), lack of details of the methods, no signs of reproducibility, and inconsistent experimentation and design motivations.




**Summary Of The Review:**

Poor presentation and quality. While novelty exists in the motivation, it is executed in an unscientific fashion. It seems like a paper thrown all together at the last moment and I do not recommend publishing of the paper even after significant registrations.

--------------------
Post Rebuttal: Given the need for a significant revision, I recommend rejecting the paper from ICLR 2023 but encourage the authors to make the material more accessible to readers in the next iteration.

---

> ### Author Response · Authors · 2022-11-12
> **Response to Reviewer uiqw**
>
> We thank reviewer uiqw for his/her reasonable criticism.
>
> Q1: The paper is extremely incoherent and it took me over 5 hrs to digest the information. For instance, (1) the abstract is extremely uninformative, so is the introduction, (2) some terms are used without any explanation (e.g., the word “coordinate”), (3) the word “fabulous” is confusing, (4) table, formula and figure are poorly organized and presented.
>
> A1: We sincerely apologize for the poor writing. We will rewrite the paper. For some conceptions you feel confused about we would like to provide a detailed explanation here.
>
> We call a certain distribution “Fabulous” since we observe it has a straightforward relation to the redundancy in the neural network model. The single center of the distribution reveals the position where redundant parameters concentrate. We prune away the parameters near this center to maximally remove the redundancy. However, the redundant parameters will emerge again during the retraining process, which, from the perspective of distribution analysis, appears to reconcentrate at the original center. This reconcentration accompanies the retraining process until there is nearly no redundancy in the model.
>
> The word “coordinate” actually refers to “coordinate system”. We view previous pruning criteria (such as L1-norm of weights/filters) as the coordinate systems on which we can conduct distribution analysis.
>
> Q2: The motivation for the design principles come from unstructured pruning and the paper experiments on structured pruning, I am not sure how that works.
>
> A2: The focus of this work is not the pruning at a certain dimension (e.g., filter-/channel-wise), but the automation of designing pruning criteria. Our Crossword Puzzle search engine can also search for a criterion that is suitable for unstructured pruning. But we only display the best searching outcome, SFN, which happens to be at filter-wise.
>
> Q3: I have no idea how the candidates for coordinates are even generated for search.
>
> A3: As mentioned in Section 4, our candidates are the linear combinations of multiple basic pruning criteria. If we have “L1-norm of the filter” ($\|F\|_1$) and “L2-norm of the filter” ($\|F\|_2$) in our search space, we will generate candidates like $\|F\|_1$, $\|F\|_2$, $\|F\|_1 + \|F\|_2$ and $\|F\|_1 - \|F\|_2$. We realize that this paper lacks a clear description of the searching engine. We will provide more details in the next version.
>
> Q4: Figure 5 shows your method is more inaccurate than Han et al., I do not understand.
>
> A4: Deep Compression is an unstructured pruning method, which might outperform the filter-wise methods (our SFN) in some cases.
>
> Q5: The experiments are performed on limited evaluations and at 50% structured sparsity which is not even considered to be sparse because of the known redundancies. The usual comparisons happen at over 70% for ImageNet.
>
> A5: We don’t show the results for 70% pruned models for the coherence of comparison (some works we compare to don’t report their results at 70%). We will display related results in the next version.
>
> Q6: FLOPS is not directly proportional to sparsity due to non-uniformity across layers, so that metric needs to be fixed.
>
> A6: Thanks for your suggestions. We will use a more suitable metric, like the actual inference speedup, to measure our method.

---

> > ### Comment · Reviewer_uiqw · 2022-11-12
> > **Thanks for the response**
> >
> > Dear Authors,
> >
> > Thanks for the rebuttal. In light of your response, it is clear the paper needs a significant overhaul and is hence not publishable in its current form. I wish the authors the best for their next revision.
> >
> > Thanks.

---

### Decision · Program_Chairs · 2023-01-20

**Decision:**

Reject

**Justification For Why Not Higher Score:**

This paper did not receive a higher score because the reviewers were unable to evaluate it on its technical merits due to poor writing.

**Justification For Why Not Lower Score:**

N/A

**Metareview: Summary, Strengths And Weaknesses:**

This paper was so difficult for the reviewers to understand that they could not make an adequate technical evaluation of the paper. I recommend rejection and encourage the authors to focus on writing so that the paper can get a full technical evaluation in a future submission.